# OpenReview forum: "TimeBridge: Non-Stationarity Matters for Long-term Time Series Forecasting"
_ICML.cc/2025/Conference — ICML 2025 poster_

### Official Review · Reviewer_yRZB · 2025-03-10

**Overall Recommendation:** 4

**Summary:**

This paper proposes a novel long-short-term representation modeling approach for handling non-stationarity in multivariate time series. For short-term representation, the integrated attention mechanism is employed to model temporal dependencies. For long-term representation, the cointegrated attention mechanism captures inter-channel dependencies while preserving cointegration relationships. Overall, it provides a novel perspective on modeling non-stationary multivariate time series.

**Claims And Evidence:**

This claim aligns well with classical time series analysis and is supported by existing literature.

**Essential References Not Discussed:**

The literature review is thorough.

**Experimental Designs Or Analyses:**

The experiments are well-designed, covering standard datasets and baselines. However, the choice of loss function (L1 loss) warrants further analysis, as it may involve unfair comparisons.

**Methods And Evaluation Criteria:**

The experimental setup is comprehensive and fair, covering mainstream datasets and models in multivariate time series forecasting.

**Other Comments Or Suggestions:**

The font size in Table 17 could be increased for better readability.

**Other Strengths And Weaknesses:**

**Strengths**: While traditional theories on non-stationarity are well-established, many recent deep learning-based approaches either overlook or only partially incorporate them, limiting their effectiveness. This paper successfully integrates these classical principles into deep learning models, demonstrating strong empirical performance, which provides valuable insights for the field.

**Weaknesses**: As mentioned, the loss function design (L1 loss) lacks sufficient discussion on its contribution to performance improvements.

**Questions For Authors:**

Could the authors provide further insights into the rationale behind their loss function choice and its comparative effectiveness? If all methods were evaluated under the same loss function, would the proposed approach still achieve significant improvements? Clarifying this point is crucial to distinguishing whether the performance gains stem from handling non-stationarity or from the loss function itself.

**Relation To Broader Scientific Literature:**

The paper's motivation aligns well with classical time series theories. These concepts have been well-established in prior research.

**Theoretical Claims:**

No relevant theoretical claims are made in this paper.

---

> ### Author Rebuttal · Authors · 2025-04-01
>
> Thanks for your constructive feedback and valuable insights into our work. Below, we address your concerns:
>
> **Q1:** The loss function lacks sufficient discussion on its contribution to performance improvements.
>
> **A1:**
> We conducted additional experiments where TimeBridge was trained using the **MSE loss**, denoted as "TimeBridge (MSE loss)", and compared the results with the best baseline, DeformableTST. Regardless of the loss function, TimeBridge outperforms DeformableTST in most datasets (See Table A), highlighting the effectiveness of our framework. We will incorporate the full results in the revision of our paper.
>
> **Table A: Average results of different prediction length {$96, 192, 336, 720$} with MSE Loss and Hybrid MAE Loss**
> ||TimeBridge|TimeBridge (MSE loss)|DeformableTST|
> |-|-|-|-|
> |ETTm1|**0.344**/**0.379**|0.353/0.388|0.348/0.383|
> |ETTm2|**0.246**/**0.310**|**0.246**/**0.310**|0.257/0.319|
> |ETTh1|**0.397**/0.424|0.399/**0.420**|0.404/0.423|
> |ETTh2|0.341/0.382|0.346/0.394|**0.328**/**0.377**|
> |Weather|0.219/**0.249**|**0.218**/0.250|0.222/0.262|
> |Electricity|**0.149**/**0.245**|**0.149**/0.246|0.161/0.261|
> |Traffic|**0.360**/0.255|**0.360**/**0.252**|0.391/0.278|
> |Solar|**0.181**/0.239|**0.181**/**0.238**|0.185/0.254|
> |Climate|1.057/0.494|**1.052**/**0.486**|1.060/0.496|
>
>
> **Q2:** The font size in Table 17 could be increased for better readability.
>
> **A2:**
> Thank you for the suggestion. We will adjust the font size of the large tables in the revised version to enhance readability.
>
> **Q3:** Could the authors provide further insights into the rationale behind their loss function choice and its comparative effectiveness? If all methods were evaluated under the same loss function, would the proposed approach still achieve significant improvements? Clarifying this point is crucial to distinguishing whether the performance gains stem from handling non-stationarity or from the loss function itself.
>
> **A3:**
> From Table A above, it is clear that the hybrid MAE loss improves TimeBridge's performance on four ETT datasets. The ETT datasets have relatively few channels ($C=7$), which limits TimeBridge’s ability to utilize the Cointegrated Attention module to capture cointegration information. In fact, we only use the Integrated Attention to model ETT datasets (see Table 8 in our paper). Additionally, the **ETT data exhibits a high degree of random fluctuation, which is better modeled using a loss function that weighs both time and frequency components.** Hence, the hybrid MAE loss strengthens the modeling of short-term dependencies in such datasets. On datasets with more channels (e.g., Electricity, Traffic, Solar), the choice of loss function has minimal impact, as both losses yield similar results. We will incorporate this analysis in the revised version of our manuscript.
>
>
> We are deeply grateful for your recognition of our work and your constructive feedback. If you have any further questions or suggestions, we would be more than happy to address them.

---

> > ### Comment · Reviewer_yRZB · 2025-04-04
> >
> > Thanks for your new results. My concerns have been addressed, and I will maintain my original score.

---

> > > ### Author Response · Authors · 2025-04-04
> > >
> > > Thank you again for your valuable comments and for acknowledging the additional experimental results. We truly appreciate your positive recognition of our work.

---

### Official Review · Reviewer_2KKn · 2025-03-10

**Overall Recommendation:** 4

**Summary:**

This study focuses on the problem of time series forecasting (TSF) by addressing the dual challenge of integrating both long-term and short-term relationships. The proposed TimeBridge employs a dual-attention framework: initial patch-level integrated attention for short-term pattern analysis and cointegrated attention to model long-term temporal correlations. The practical performance of TimeBridge has been validated across various real-world datasets.

**Claims And Evidence:**

The claim aligns with previous work in TSF.

**Essential References Not Discussed:**

The most relevant works have already been encompassed.

**Experimental Designs Or Analyses:**

The experimental design adheres to the most widely adopted baselines in TSF, ensuring the fairness.

**Methods And Evaluation Criteria:**

The experimental datasets and evaluation metrics employed in this study strictly follow the most prevalent standards established in the TSF field.

**Other Comments Or Suggestions:**

While the study represents an effort in modeling cointegration through deep learning, the architectural design of the proposed modules appears relatively simplistic and could benefit from further sophistication.

**Other Strengths And Weaknesses:**

Strengths:

1. The paper is well-motivated. It introduces a novel approach and a new perspective from classical time series analysis for addressing non-stationary time series data.

2. This paper is well written, with well-defined mathematical notations.

3. The experiment is comprehensive, featuring extensive ablation studies that systematically validate the contribution of each component.

Weaknesses:

1. As indicated in Table 6, the sequential selection strategy of CI and CD significantly impacts prediction performance on Electricity and Traffic datasets, yet this phenomenon remains insufficiently explained in the text.

2. The manuscript lacks computational complexity analysis, making it difficult to assess the model's efficiency and memory requirements.

**Questions For Authors:**

I am particularly interested in whether the financial experiments have been implemented in real-world trading scenarios? Could the authors provide some insights into the practical deployment?

**Relation To Broader Scientific Literature:**

Non-stationarity represents a crucial aspect in series modeling, and this article possesses the potential to exert a broader impact on related scientific endeavors.

**Theoretical Claims:**

No theoretical claims.

---

> ### Author Rebuttal · Authors · 2025-04-01
>
> Thank you for your feedback and insightful comments on our work. Below, we address each of your concerns and questions:
>
> **Q1:** As indicated in Table 6, the sequential selection strategy of CI and CD significantly impacts prediction performance on Electricity and Traffic datasets, yet this phenomenon remains insufficiently explained in the text.
>
> **A1:** We believe that modeling short-term dependencies first is crucial because these local temporal features are essential for accurate short-term forecasting. **By using Integrated Attention (CI) to capture short-term dependencies before Cointegrated Attention (CD) models long-term relationships, we ensure that important immediate patterns are not lost, which significantly improves performance.** A similar "local first, global later" modeling strategy is also used in other works [1].
>
> **Q2:** The manuscript lacks computational complexity analysis, making it difficult to assess the model's efficiency and memory requirements.
>
> **A2:**
> We provide a detailed comparison of the theoretical complexity of our method with other mainstream Transformer-based models. The table below summarizes the theoretical computational complexity, where $C$ is the number of channels, $I$ and $O$ are the input and output lengths, and $S$ is the patch length:
> |TimeBridge|iTransformer|PatchTST|Crossformer|FEDformer|
> |-|-|-|-|-|
> |$O(\text{max}(C·(\frac{I}{S})^2，C^{3/2}))$|$O(C^2)$|$O(C·(\frac{I}{S})^2)$|$O(\frac{C}{S^2}O(I+O))$|$O(I/2+O)$|
>
> Moreover, theoretical complexity alone cannot fully capture real-world performance due to implementation differences. We tested on 2 NVIDIA 3090 GPUs, measuring training (1 epoch) and inference times for three datasets of increasing size, with results averaged over 5 runs. The results in the table below indicate that TimeBridge outperforms most Transformer-based models.
>
> |||TimeBridge|iTransformer|PatchTST|Crossformer|FEDformer|
> |-|-|-|-|-|-|-|
> |ETTm1 ($C=7$)|Training Time|72s|65s|73s|280s|449s|
> ||Inference Time|33s|31s|39s|48s|62s|
> |Electricity ($C=321$)|Training Time|252s|89s|450s|328s|510s|
> ||Inference Time|125s|78s|141s|116s|130s|
> |Traffic ($C=862$)|Training Time|409s|175s|649s|360s|485s|
> ||Inference Time|175s|154s|207s|171s|196s|
>
>
> **Q3:** While the study represents an effort in modeling cointegration through deep learning, the architectural design of the proposed modules appears relatively simplistic and could benefit from further sophistication.
>
> **A3:**
> We would like to clarify that our method was **intentionally designed with a simple, intuitive structure to address the challenges of non-stationarity and dependency modeling effectively**:
> - **Integrated Attention** focuses on capturing short-term dependencies while mitigating spurious regressions, using a streamlined approach that normalizes the attention map to remove non-stationary components.
> - **Cointegrated Attention** preserves non-stationarity to model long-term cointegration relationships between variables.
> - **Patch Downsampling** bridges the two blocks, allowing the Integrated Attention block to process aggregated short-term information, which feeds into the Cointegrated Attention block for long-term modeling.
>
> We further provide a theoretical analysis of the modeling rationale behind these design choices, which can be found in Reviewer 1BTh’s Answer 1 (**A1**).
>
>
> **Q4:** I am particularly interested in whether the financial experiments have been implemented in real-world trading scenarios? Could the authors provide some insights into the practical deployment?
>
> **A4:** Thank you for your interest in the practical deployment of our model. Our financial experiments were conducted using the **Qlib** [2] platform for historical backtesting, with simulated trading costs to approximate real-world scenarios. While real-world deployment would need to address challenges such as latency and liquidity, we have not yet integrated with live trading systems due to data privacy and regulatory constraints.
>
>
> [1] MICN: Multi-scale Local and Global Context Modeling for Long-term Series Forecasting  (ICLR 2023)
>
> [2] https://github.com/microsoft/qlib
>
>
> Thank you for your valuable feedback. If you have any further questions or need clarification, feel free to follow up.

---

> > ### Comment · Reviewer_2KKn · 2025-04-03
> >
> > I appreciate the authors for thoroughly addressing the previous concerns. Given the financial focus of this work, I would like to request further clarification. The paper states that cointegration relationships are modeled across stocks. Could the authors explicitly clarify which aspect of the stocks (e.g., raw price trajectories, log-returns, or other derived signals) the cointegration mechanism primarily operates on?

---

> > > ### Author Response · Authors · 2025-04-04
> > >
> > > We sincerely appreciate the reviewer’s acknowledgment of our previous responses. The core focus of our cointegration modeling is the raw price trajectories of stocks, **as their non-stationary nature aligns with the prerequisites for cointegration analysis. Long-term equilibrium relationships between industries (e.g., the price level connection between the automobile and steel industries in cost-linked price movements) are directly reflected in price trends, rather than in short-term return fluctuations.** By analyzing the raw price series, we can effectively capture the co-movement patterns along the upstream and downstream supply chain, providing a basis for arbitrage strategies. In contrast, stationary return series would not yield cointegration relationships with meaningful economic interpretation. We will include this analysis in the revised version.

---

### Official Review · Reviewer_CeE3 · 2025-03-13

**Overall Recommendation:** 3

**Summary:**

This paper introduces TimeBridge, a methodological framework addressing non-stationarity in long-term time series forecasting. The proposed approach is structured around two core mechanisms: Integrated Attention, which seeks to mitigate short-term non-stationarity by stabilizing localized variations, and Cointegrated Attention, designed to retain and model long-term non-stationary dependencies. Empirical evaluations across multiple benchmark datasets suggest that TimeBridge performs competitively relative to state-of-the-art (SOTA) forecasting models.

## update after rebuttal

After carefully reading through all the reviewers' comments, I believe the strong performance of this work across multiple dimensions is truly commendable. Accordingly, I have raised my score, with the expectation that the authors will revise and improve the identified concerns in the camera-ready version.

**Claims And Evidence:**

The manuscript presents claims that are only partially substantiated by empirical results, and several key limitations emerge:
- The proposed framework purports to be an effective mechanism for handling non-stationarity; however, it omits direct comparisons with seasonal-trend decomposition methods [1][2], which are widely recognized as robust techniques for handling non-stationary components in time series data. Moreover, the definitions of stationary and non-stationary in the paper seem to be mere rebranding of trend and seasonal components.
- The methodological novelty of the work appears to be constrained. The proposed architecture largely constitutes a reconfiguration of existing methodologies [3][4][5][6], rather than introducing a fundamentally novel paradigm.
- While the reported experimental results indicate performance improvements, the magnitude of these improvements is relatively modest and does not conclusively establish the superiority of TimeBridge over existing approaches.

[1]  Autoformer: Decomposition Transformers with Auto-Correlation for Long-Term Series Forecasting.

[2] First De-Trend then Attend: Rethinking Attention for Time-Series Forecasting.

[3] Crossformer: Transformer Utilizing Cross-Dimension Dependency for Multivariate Time Series Forecasting

[4] Fredformer: Frequency Debiased Transformer for Time Series Forecasting

[5] A Time Series is Worth 64 Words: Long-term Forecasting with Transformers

[6] UniTST: Effectively Modeling Inter-Series and Intra-Series Dependencies for Multivariate Time Series Forecasting

**Essential References Not Discussed:**

Yes, the study should provide an in-depth discussion of:

- **Seasonal-trend decomposition techniques** to contextualize the proposed approach within established non-stationarity handling frameworks.
- **Recent advancements in hybrid modeling approaches**, which also address long-term dependencies in time series forecasting.

**Experimental Designs Or Analyses:**

The experimental design and results were examined, revealing the following limitations:
- The reported performance gains are insufficiently substantial to position TimeBridge as a transformative advancement over existing methodologies.

**Methods And Evaluation Criteria:**

The selection of benchmark datasets and evaluation metrics (MSE, MAE) aligns with standard practices in long-term time series forecasting. However, the study would benefit from:
- **Explicit comparisons with seasonal-trend decomposition methods**, given their proven efficacy in handling non-stationarity.
- A more rigorous ablation study to disentangle the individual contributions of Integrated and Cointegrated Attention, ensuring a precise understanding of their respective roles in mitigating non-stationarity.

**Other Comments Or Suggestions:**

- The theoretical grounding of Integrated and Cointegrated Attention should be more explicitly justified within the context of existing time series modeling literature.
- The ablation study should provide greater granularity in assessing the distinct contributions of each proposed component.

**Other Strengths And Weaknesses:**

- **Strengths**: The paper correctly identifies non-stationarity as a critical issue and attempts to address both short-term and long-term dependencies within a unified forecasting framework.
- **Weaknesses**: The degree of methodological novelty is limited, as the proposed framework appears to primarily constitute a recombination of existing techniques rather than a fundamentally new paradigm. Furthermore, the lack of direct comparisons with seasonal-trend decomposition models significantly undermines the empirical validation of the proposed approach.

**Questions For Authors:**

1. How does TimeBridge substantively differentiate itself from existing attention-based time series forecasting models?
2. Would a decomposition-based or frequency-domain approach offer comparable performance with potentially greater computational efficiency?

**Relation To Broader Scientific Literature:**

The study contributes to time series forecasting by integrating non-stationarity mitigation techniques into an attention-based framework. However, several key concerns remain unaddressed:
- Seasonal-trend decomposition methods have long been established as effective solutions for handling non-stationarity, yet the paper does not explicitly contrast its approach with these techniques.
- The methodology and structure of TimeBridge bear strong resemblance to several prior works, including [3][4][5][6]. The extent to which TimeBridge provides a substantive departure from these works remains unclear.

[3] Crossformer: Transformer Utilizing Cross-Dimension Dependency for Multivariate Time Series Forecasting

[4] Fredformer: Frequency Debiased Transformer for Time Series Forecasting

[5] A Time Series is Worth 64 Words: Long-term Forecasting with Transformers

[6] UniTST: Effectively Modeling Inter-Series and Intra-Series Dependencies for Multivariate Time Series Forecasting

**Theoretical Claims:**

The manuscript does not provide formal mathematical proofs.

Although it references cointegration principles, it lacks a rigorous theoretical justification for why the proposed attention-based mechanisms offer an optimal means of capturing non-stationary dependencies. The absence of a formal derivation raises concerns regarding the theoretical soundness of the proposed approach.

---

> ### Author Rebuttal · Authors · 2025-04-01
>
> Thank you for your feedback and insightful comments on our work. Below, we address your questions:
>
> **Q1:** It omits direct comparisons with seasonal-trend decomposition methods.
>
> **A1:** We add comparison with seasonal-trend decomposition methods (Autoformer [1] and TDformer [2]). The table below shows the average prediction results across all datasets. Full results will be included in the revised paper.
>
> ||TimeBridge|Autoformer|TDformer|
> |-|-|-|-|
> ||MSE/MAE|MSE/MAE|MSE/MAE|
> |ETTm1|**0.344**/**0.379**|0.588/0.517|0.380/0.406|
> |ETTm2|**0.246**/**0.310**|0.324/0.368|0.267/0.325|
> |ETTh1|**0.397**/**0.424**|0.496/0.487|0.486/0.479|
> |ETTh2|**0.341**/**0.382**|0.453/0.462|0.378/0.411|
> |Weather|**0.219**/**0.249**|0.338/0.382|0.236/0.271|
> |Electricity|**0.149**/**0.245**|0.227/0.338|0.177/0.282|
> |Traffic|**0.360**/**0.255**|0.628/0.379|0.570/0.322|
> |Solar|**0.181**/**0.239**|0.340/0.380|0.208/0.251|
> |Climate|**1.057**/**0.494**|1.456/0.584|1.364/0.626|
>
> **Q2:** The definitions of stationary and non-stationary in the paper seem to be mere rebranding of trend and seasonal components.
>
> **A2:** While trend and seasonal components can be viewed as corresponding to non-stationary and stationary elements, previous works have mainly focused on modeling them within individual channels. They haven’t fully addressed the need to preserve non-stationarity across channels to capture long-term cointegration. **Our paper’s core insight is that short-term dependencies can be modeled within stationary channels, while long-term cointegration requires preserving the non-stationary relationships between variables.** Thus, we analyze non-stationarity’s distinct impacts on both intra- and inter-variable modeling, rather than just decomposing the sequence within channels as previous methods [1,2] have done.
>
> **Q3:** The improvements are modest and don't definitively prove TimeBridge's superiority over existing methods.
>
> **A3:** As shown in **Table 1**, TimeBridge consistently achieves superior performance in long-term forecasting, reducing MSE and MAE by 1.85%/2.49%, 5.56%/4.12%, and 13.66%/7.58% compared to DeformableTST, ModernTCN, and TimeMixer, respectively. Additionally, our model demonstrates exceptional performance in financial forecasting (**Table 3**), highlighting its ability to capture complex cointegration relationships in financial markets. These results confirm the effectiveness of our approach.
>
> **Q4:** The methodology and structure of TimeBridge bear strong resemblance to several prior works, including [3,4,5,6].
>
> **A4:** We design our model with a unique focus on non-stationarity in both short-term stability and long-term dependencies. **Integrated Attention stabilizes short-term dependencies by removing non-stationary features from the Query-Key pairs, while Cointegrated Attention emphasizes the necessity of preserving non-stationarity for modeling long-term cointegration relationships between channels.**
> In contrast, PatchTST [5] does not address non-stationarity and focuses solely on the temporal dimension, missing the rich cointegration relationships between variables. Crossformer [3] and UniTST [6] model short-term dependencies across and within channels but are theoretically susceptible to spurious regressions and lack explicit consideration of long-term cointegration. Fredformer [4], though utilizing frequency-domain attention to capture long-term signals, struggles with fine-grained temporal trends in the time domain, limiting its ability to model short-term dependencies effectively.
>
> **Q5:** A more detailed ablation study is needed to clarify the individual contributions of Integrated and Cointegrated Attention in mitigating non-stationarity.
>
> **A5:** We have conducted an ablation study on the impact and order of Integrated Attention and Cointegrated Attention in **Table 5** in our paper. The results indicate that both attention mechanisms must be used together, with Integrated Attention applied first, followed by Cointegrated Attention.
>
> **Q6:** The theoretical grounding of Integrated and Cointegrated Attention should be more explicitly justified within the context of existing time series modeling literature.
>
> **A6:** We have provided a more explicit theoretical justification for Integrated and Cointegrated Attention in Reviewer 1BTh’s Answer 1 (**A1**).
>
> **Q7:** Would a decomposition-based or frequency-domain approach offer comparable performance with potentially greater computational efficiency?
>
> **A7:** We compared TimeBridge with methods such as TimeMixer (decomposition-based), TimesNet (frequency-domain), and the newly added Autoformer and TDformer in **A1**, all of which demonstrate that TimeBridge consistently outperforms them. While decomposition-based or frequency-domain methods may offer comparable performance, we currently do not have conclusive evidence to support this.
>
> We appreciate your insightful feedback. If you have any further questions or need clarification, feel free to follow up.

---

> > ### Comment · Reviewer_CeE3 · 2025-04-05
> >
> > Thank you for your detailed rebuttal. While most of my concerns have been addressed, a few points remain:
> >
> > 1. Regarding Q1: TDformer and Autoformer are not the state-of-the-art STD-based methods. I recommend you compare and include Leddam [1] (ICML2024), which employs a similar Channel-Independent (CI) and Channel-Dependent (CD) design as your work.
> >
> > 2. Regarding Q3: I noticed that you use a loss function different from other baselines—switching from the conventional MSE to MAE and incorporating both time and frequency domain information. In my reproduction, I observed that this loss function significantly impacts the results. **However, the paper does not describe this loss function in detail or provide ablation experiments to isolate its effect.** This raises concerns that the observed improvements might largely stem from using a different loss function. I strongly recommend that you compare TimeBridge, PatchTST, ModernTCN, and Leddam under both the traditional MSE loss and the loss function you employ, along with a detailed explanation of its necessity and benefits.
> >
> > 3. Regarding Q6: **I strongly suggest incorporating the theoretical discussion on Integrated and Cointegrated Attention into the main text** to clarify its grounding in existing time series modeling literature.
> >
> > **I am confident that the authors can address the aforementioned issues in the final version.**
> >
> > After carefully reading through all the reviewers' comments, I believe the strong performance of this work across multiple dimensions is truly commendable. Accordingly, **I have raised my score, with the expectation that the authors will revise and improve the identified concerns in the camera-ready version.**
> >
> > *A minor suggestion regarding Reviewer 2KKn's Q.1 on the justification of the CI and CD strategies:*
> >
> > I recommend referring to **“The Capacity and Robustness Trade-off”** [2] (TKDE), which offers valuable theoretical insights relevant to your discussion.
> >
> > One of the key findings of that work is as follows:
> > ***The Channel Dependent (CD) strategy exhibits high capacity but low robustness, whereas the Channel Independent (CI) strategy has lower capacity but higher robustness. In many real-world, non-stationary time series characterized by distribution drifts, robustness tends to be more critical than capacity for achieving reliable forecasting performance. As a result, the CI strategy often outperforms the CD strategy in practice.***
> >
> > Incorporating a more detailed discussion of this Trade-off in the main paper could significantly strengthen your justification and highlight the robustness of your method under real-world conditions.
> >
> > *[1] Revitalizing Multivariate Time Series Forecasting: Learnable Decomposition with Inter-Series Dependencies and Intra-Series Variations Modeling.*
> >
> > *[2] The Capacity and Robustness Trade-off: Revisiting the Channel
> > Independent Strategy for Multivariate Time Series Forecasting.*

---

> > > ### Author Response · Authors · 2025-04-06
> > >
> > > We are immensely grateful for your positive recognition of our work and the additional experimental results. Below are our responses to your new points:
> > >
> > > ---
> > >
> > > **Q1:** Compare and include Leddam.
> > >
> > > **R1:** Thank you for your suggestion. We compared TimeBridge with Leddam across four forecasting horizons, as shown in the table below. **Using the hyperparameter search strategy in Appendix E.1, we found that TimeBridge consistently outperforms Leddam, especially on datasets with more channels.** Additionally, our search results for Leddam are better than those reported in their paper for a fixed input length of 96, which aligns with their observation that extending the look-back window improves forecasting performance. We will include the complete results in the revised version.
> > >
> > >
> > > ||TimeBridge|Leddam|
> > > |-|-|-|
> > > ||MSE/MAE|MSE/MAE|
> > > |ETTm1|**0.344**/**0.379**|0.354/0.381|
> > > |ETTm2|**0.246**/**0.310**|0.265/0.320|
> > > |ETTh1|**0.397**/**0.424**|0.415/0.430|
> > > |ETTh2|**0.341**/**0.382**|0.345/0.391|
> > > |Weather|**0.219**/**0.249**|0.226/0.264|
> > > |Electricity|**0.149**/**0.245**|0.162/0.256|
> > > |Traffic|**0.360**/**0.255**|0.452/0.283|
> > > |Solar|**0.181**/**0.239**|0.223/0.264|
> > >
> > > ---
> > >
> > > **Q2:** Discussion of Loss function.
> > >
> > > **R2:** We compared TimeBridge with the MSE loss in Reviewer yRZB’s **A1**, showing it still outperforms the best baseline, DeformableTST. As detailed in Reviewer yRZB’s **A3**, **the hybrid MAE loss primarily enhances performance on the ETT dataset, where the limited channels hinder cointegration modeling.** Given the high degree of random fluctuation in ETT data, using a loss function that combines time and frequency weighting better captures its dynamics. **For datasets with more channels (e.g., Traffic and Solar), the impact of the loss function is minimal.** We will further include results for other baselines using this loss function.
> > >
> > > ---
> > >
> > > **Q3:** Theoretical discussion on Integrated and Cointegrated Attention.
> > >
> > > **R3:** We will provide a detailed theoretical discussion of the impact of Integrated and Cointegrated Attention on non-stationarity in the revised version.
> > >
> > > ---
> > >
> > > **Q4:** Refer to “The Capacity and Robustness Trade-off” to justify the CI and CD strategies.
> > >
> > > **R4:** We will refer to the findings in “The Capacity and Robustness Trade-off” to enrich the discussion of CI and CD strategies in the subsection on **Ablation on Integrated and Cointegrated Attention impact and order.**
> > >
> > > ---
> > >
> > > Guided by your suggestions, we have deepened our understanding of the critical issues you highlighted, significantly improving the quality of our work. Thank you once again for your valuable feedback.

---

### Official Review · Reviewer_1BTh · 2025-03-14

**Overall Recommendation:** 3

**Summary:**

This paper introduces a new framework to tackle the challenges posed by non-stationarity in multivariate time series forecasting. It addresses both short-term fluctuations and long-term trends by employing two specialized attention mechanisms, i.e., Integrated Attention and Cointegrated Attention. The framework is validated with extensive experiments on real-world datasets, demonstrating superior performance in both short-term and long-term forecasting.

**Claims And Evidence:**

Yes

**Essential References Not Discussed:**

No

**Experimental Designs Or Analyses:**

The overall experimental designs look great.

**Methods And Evaluation Criteria:**

Yes

**Other Comments Or Suggestions:**

See above

**Other Strengths And Weaknesses:**

Strengths:
1. Addressing the dual challenge of non-stationarity in short-term stability and long-term dependencies is interesting, as traditional methods either neglect or insufficiently manage these aspects.
2. The framework has been rigorously tested across multiple datasets, demonstrating consistent state-of-the-art performance. The inclusion of financial forecasting datasets enhances the credibility of the results.


Weaknesses:
1. While the authors criticize some existing methods for incorporating non-stationary factors without a robust theoretical framework, the proposed method in this paper also lacks a rigorous theoretical foundation. A more detailed theoretical exploration could strengthen the framework's validity.
2. The novelty of the proposed work could be further justified. The use of attention mechanisms to manage correlations between patches or among variates is not entirely novel, as similar approaches have been explored in channel-dependent transformers. Highlighting distinct advantages or improvements over these methods would clarify the contribution of this work.
3. In short-term forecasting, where ARIMA is recognized as a strong baseline method, the authors might benefit from discussing its relative performance. This comparison could provide a clearer benchmark for evaluating the advantages of the proposed framework.

**Questions For Authors:**

See above

**Relation To Broader Scientific Literature:**

The authors have discussed the related work in Normalization and Dependecy Modeling in time series forecasting.

**Theoretical Claims:**

Not applicable

---

> ### Author Rebuttal · Authors · 2025-04-01
>
> Thanks for your insightful comments on our work. Below, we address each of your questions:
>
> **Q1:** A more detailed theoretical exploration could strengthen the framework's validity.
>
> **A1:** Thank you for your valuable comments. Below we provide a concise theoretical explanation grounded in classical stochastic processes, specifically Brownian motion, to clarify the rationale behind our design.
>
> **Proposition 1: Spurious Attention from Non-Stationary Inputs**
>
> Consider a standard Brownian motion $X_t \sim I(1)$, where:
>
> $X_t = X_{t-1} + u_t,\quad u_t \sim \mathcal{N}(0, \sigma^2)$.
>
> Then we have:
>
> $\text{Mean}(X_t)=0, \quad \text{Var}(X_t) = t\sigma^2,\quad \text{Cov}(X_{t_1}, X_{t_2}) = \min(t_1, t_2)\sigma^2$
>
> Let two input patches of length $S$ be:
>
> $p_i = [X_{t+i+1}, \dots, X_{t+i+S}],\quad p_j = [X_{t+j+1}, \dots, X_{t+j+S}]$
>
> Their attention score $\text{score}(p_i, p_j)$ is approximated as:
>
> $\text{score}(p_i, p_j) \propto p_i p_j^T \propto \sum_{s=1}^{S} (X_{t+i+s} X_{t+j+s}) \propto \sum_{s=1}^{S} \text{Cov}(X_{t+i+s}, X_{t+j+s})$
>
> $
> \propto \sigma^2 \left(S\min(i, j) + \frac{S^2 + 2St + S}{2}\right)
> $
>
> This score grows with both the time index $t$ and the square of the patch length $S$, leading to spurious attention caused by global trends rather than genuine short-term dependencies. **As shown in Figure 4(a), many patches tend to exhibit high attention scores.**
>
> To mitigate this, we perform patch-wise detrending:
>
> $p_i' = \text{Detrend}(p_i) = [\Delta X_{t+i}, \dots, \Delta X_{t+i+S}] \sim I(0), \quad \Delta X_t = X_t - X_{t-1} $
>
> In this case, the variance becomes stable:
>
> $\text{Var}(\Delta X_t) = \sigma^2,\quad \text{score}(p_i', p_j') \propto S\sigma^2$
>
> This makes the attention mechanism focus on true short-term patterns, unaffected by long-term drift.
>
> ---
> **Proposition 2: Importance of Non-Stationarity in Capturing Cointegration**
>
> Let $X_t, Y_t \sim I(1)$ be two non-stationary time series with a cointegration relationship:
>
> $Z_t = X_t - \beta Y_t \sim I(0)$
>
> where $\beta$ is a constant coefficient. If we remove non-stationarity via detrending:
>
> $\Delta X_t = \text{Detrend}(X_t),\quad \Delta Y_t = \text{Detrend}(Y_t)$
>
> Then:
>
> $
> Z_t = \Delta X_t - \beta \Delta Y_t = \epsilon_t
> $
>
> which becomes a **random noise sequence**. This destroys the cointegration signal, making it impossible for attention mechanisms to capture long-term equilibrium relationships. **Figure 4(b) illustrates that removing non-stationary components eliminates the vast majority of cointegration information.**
>
> ---
> These two propositions together explain our architectural choice:
> - For **short-term modeling**, we eliminate non-stationarity to avoid spurious regressions.
> - For **long-term modeling**, we preserve non-stationarity to retain meaningful cointegration.
>
> We will incorporate this theoretical analysis in greater detail in the revision of our paper.
>
> **Q2:** Highlighting distinct advantages or improvements over these methods would clarify the contribution of this work.
>
> **A2:** **We design our model by focusing on the distinct characteristics of non-stationary data, specifically addressing short-term stability and long-term dependencies.** This insight leads to the adoption of simple yet effective strategies that align with the nature of non-stationary data. Unlike other attention methods that directly model short-term dependencies and are prone to spurious regressions, Integrated Attention stabilizes short-term modeling by removing non-stationary features from the Query-Key pairs. In contrast, Cointegrated Attention emphasizes the necessity of retaining non-stationary features for capturing long-term cointegration relationships, setting it apart from recent channel-dependent Transformer models. These models either fail to consider long-term cointegration [1] or overlook it entirely [2].
>
> [1] Crossformer: Transformer Utilizing Cross-Dimension Dependency for Multivariate Time Series Forecasting
>
> [2] Revitalizing Multivariate Time Series Forecasting: Learnable Decomposition with Inter-Series Dependencies and Intra-Series Variations Modeling
>
> **Q3:** ARIMA is a strong baseline for short-term forecasting; the authors should compare its performance to highlight the advantages of their framework.
>
> **A3:** Since ARIMA is for univariate forecasting and our PeMS datasets are multivariate, we compared TimeBridge with VARIMA, which is more appropriate for multivariate forecasting. The table below shows that TimeBridge outperforms VARIMA in most cases. This comparison will be updated in the revised paper.
> ||TimeBridge|VARIMA|
> |-|-|-|
> ||MAE/MAPE/RMSE|MAE/MAPE/RMSE|
> |PeMS03|14.52/**14.21**/23.10|**13.78**/17.95/**19.95**|
> |PeMS04|**19.24**/**12.42**/**31.12**|24.87/15.61/36.26|
> |PeMS07|**20.43**/**8.42**/**33.44**|26.00/11.21/37.67|
> |PeMS08|**14.98**/**9.56**/**23.77**|19.38/12.49/28.02|
>
> We sincerely appreciate your constructive feedback. If you have any further questions or need clarification, please feel free to reach out.

---

### Decision · Program_Chairs · 2025-05-01

**Decision:**

Accept (poster)

**Comment:**

This paper proposes a methodological framework addressing nonstationarity in long-term time series forecasting. The proposed approach is structured around two core mechanisms: Integrated Attention, which seeks to mitigate short-term non-stationarity by stabilizing localized variations, and Cointegrated Attention, designed to retain and model long-term non-stationary dependencies. As pointed out by the reviewers, the studied problem, addressing the dual challenge of nonstationarity in short-term stability and long-term dependencies, is interesting, and the empirical performance of the proposed approaches is impressive. A more detailed theoretical exploration of why the proposed approach works would strengthen the work.